# Mitochondrial Ca^2+^ Dynamics in MCU Knockout *C. elegans* Worms

**DOI:** 10.3390/ijms21228622

**Published:** 2020-11-16

**Authors:** Pilar Álvarez-Illera, Paloma García-Casas, Rosalba I Fonteriz, Mayte Montero, Javier Alvarez

**Affiliations:** Institute of Biology and Molecular Genetics (IBGM), Department of Biochemistry and Molecular Biology and Physiology, Faculty of Medicine, University of Valladolid and CSIC, Ramón y Cajal, 7, E-47005 Valladolid, Spain; pilar_alvill@hotmail.com (P.Á.-I.); palomabio21@gmail.com (P.G.-C.); rfonteri@ibgm.uva.es (R.IF.); mmontero@ibgm.uva.es (M.M.)

**Keywords:** *C. elegans*, mitochondria, mitochondrial calcium uniporter, MCU, knockout, calcium dynamics

## Abstract

Mitochondrial [Ca^2+^] plays an important role in the regulation of mitochondrial function, controlling ATP production and apoptosis triggered by mitochondrial Ca^2+^ overload. This regulation depends on Ca^2+^ entry into the mitochondria during cell activation processes, which is thought to occur through the mitochondrial Ca^2+^ uniporter (MCU). Here, we have studied the mitochondrial Ca^2+^ dynamics in control and MCU-defective *C. elegans* worms in vivo, by using worms expressing mitochondrially-targeted YC3.60 yellow cameleon in pharynx muscle. Our data show that the small mitochondrial Ca^2+^ oscillations that occur during normal physiological activity of the pharynx were very similar in both control and MCU-defective worms, except for some kinetic differences that could mostly be explained by changes in neuronal stimulation of the pharynx. However, direct pharynx muscle stimulation with carbachol triggered a large and prolonged increase in mitochondrial [Ca^2+^] that was much larger in control worms than in MCU-defective worms. This suggests that MCU is necessary for the fast mitochondrial Ca^2+^ uptake induced by large cell stimulations. However, low-amplitude mitochondrial Ca^2+^ oscillations occurring under more physiological conditions are independent of the MCU and use a different Ca^2+^ pathway.

## 1. Introduction

Ca^2+^ uptake by mitochondria plays multiple roles in cellular Ca^2+^ homeostasis and cell death. The increase in mitochondrial [Ca^2+^] ([Ca^2+^]_M_) activates several dehydrogenases important for energy production, thus activating the respiratory chain and increasing adenosine triphosphate (ATP) production [1,2]. In this way, [Ca^2+^]_M_ increase is a signal of cell activation and enhances energy production in order to cover the energy requirements of the response to the stimulus. Moreover, during cell activation, mitochondrial Ca^2+^ uptake constitutes a mechanism of transient cytosolic Ca^2+^ buffering that modulates cytosolic [Ca^2+^]. In particular, mitochondria are very effective taking up Ca^2+^ from local high-Ca^2+^ microdomains, such as those generated by the activation of plasma membrane or ER Ca^2+^ channels [3]. These high-Ca^2+^ cytosolic microdomains control physiological phenomena as important as neurotransmitter secretion or muscle contraction, which can therefore be modulated by mitochondria acting as local Ca^2+^ sinks. For example, mitochondrial Ca^2+^ uptake modulates Ca^2+^-dependent catecholamine secretion in chromaffin cells [4] or the propagation of cytosolic Ca^2+^ waves [5], and the close contacts between ER and mitochondria directly control Ca^2+^ transfer between both organelles [2]. As well as these roles of mitochondrial Ca^2+^ uptake in normal cell physiology, it is also widely known that [Ca^2+^]_M_ overload is a key factor controlling cell death during necrosis and apoptosis. During these processes, Ca^2+^ accumulation in mitochondria contributes to triggering of the opening of a high-conductance pore in the inner mitochondrial membrane, the so-called mitochondrial permeability transition pore, which is followed by osmotic swelling of mitochondria, rupture of the mitochondrial membranes and release of apoptotic factors to the cytosol [6,7].

The molecular substrate of the mitochondrial Ca^2+^ uniporter protein (MCU) was characterized in 2011 [8,9] and it was soon found that at least other six regulatory subunits (MCUb, MCUR1, EMRE, MICU1, MICU2 and MICU3) could be present in mammalian cells to modulate Ca^2+^ uptake though the MCU pore subunit, all of them with different and in some cases opposite functions, which are still not completely clarified [10,11,12]. Unexpectedly, in spite of the importance of mitochondrial Ca^2+^ uptake for cell physiology and the complex regulation of the uniporter, knockout mice completely lacking the MCU protein were viable and showed a very mild phenotype [13,14]. Similarly, knockout of the MCU protein in *C. elegans* has only been described to produce defects in wound closure as a result of reduced wound-induced mitochondrial superoxide production [15].

Here, we have used a knockout mutant of the MCU protein in *C. elegans* to study the effects of MCU depletion in mitochondrial Ca^2+^ dynamics. We have shown before that pharynx mitochondrial [Ca^2+^] follows the fast oscillations of cytosolic [Ca^2+^] “beat-to-beat”, which is responsible for pharynx pumping [16,17]. Now, by looking at mitochondrial [Ca^2+^] in the worm pharynx of control and *mcu-1* mutants, we show that the pattern of oscillatory Ca^2+^ dynamics in the mutants is very similar to that in the controls, although they show a smaller response to high-intensity stimulation. We conclude that MCU is essential to convey large-amplitude Ca^2+^ signals to the mitochondria. Instead, the low-amplitude [Ca^2+^]_M_ oscillations seen in pharynx muscle during normal physiological activity are not mediated by MCU activation and use a different Ca^2+^ pathway to enter mitochondria.

## 2. Results

### 2.1. Mitochondrial [Ca^2+^] Oscillations in Control and MCU-Defective Worms

We have recently used the AQ3055 strain expressing the YC3.60 Ca^2+^ sensor in the pharynx to investigate the dynamics of pharynx [Ca^2+^] in live worms under serotonin stimulation [17]. Our results showed that pharynx mitochondrial [Ca^2+^] is able to oscillate at a high frequency in vivo, reproducing the oscillations of cytosolic [Ca^2+^] and behaving in a “beat-to-beat” mode at frequencies close to 1 Hz. This indicates that the mechanisms of Ca^2+^ uptake and release from the mitochondria are fast enough to recover the resting levels after each Ca^2+^ spike.

The mitochondrial Ca^2+^ uniporter is the best known Ca^2+^ channel responsible for mitochondrial Ca^2+^ uptake, and therefore we wanted to study how that oscillatory Ca^2+^ pattern was modified in worms lacking a functional MCU protein. Surprisingly, Figure 1 shows that there were few changes in the serotonin-stimulated pharynx Ca^2+^ dynamics among controls and *mcu-1* mutants. Panels a and b of the figure show two typical 30 min Ca^2+^ records, one in the AQ3055 strain, which expresses YC3.60 targeted to the mitochondria, and the other in the *mYCmcu-1* strain, which expresses YC3.60 targeted to the mitochondria in the mutant *mcu-1* background. During the initial 15 min we monitored the spontaneous Ca^2+^ oscillations obtained in the presence of serotonin. Then, 10 mM carbachol was added to the chamber to induce a strong Ca^2+^ signal mediated by the activation of the nicotinic acethylcholine receptor. In this way, the first part of the experiment tests the transfer of small spontaneous cytosolic Ca^2+^ oscillations to the mitochondria and the second part evaluates the transfer of Ca^2+^ to the mitochondria during large cell stimulation.

Our data show that the spontaneous [Ca^2+^]_M_ oscillations were not abolished by the lack of functional MCU. Panels c and d show a magnification of a small portion of each of the records in the initial part of the experiments. The figure shows the fluorescence ratio F535/F480, and also the individual F535 and F480 fluorescence data. Both wavelengths show mirror changes, as would be expected for true changes in [Ca^2+^]_M_. Our data show that the dynamics of the [Ca^2+^]_M_ oscillations obtained in control worms or in worms lacking MCU was very similar. We analyzed the mean width and height of the Ca^2+^ oscillations, and we could only detect quite small changes in these parameters in the mutants, as seen in panel e. mYC*mcu-1* mutants showed a small but significant increase in both width and height of the [Ca^2+^]_M_ peaks.

### 2.2. Electropharyngeograms of Control and MCU-Defective Worms

The spontaneous Ca^2+^ oscillations reflect the behavior of physiological pharynx pumping, which depends on the rhythmic stimulation by cholinergic MC neurons [18]. In order to study if the changes in the kinetic parameters of the [Ca^2+^]_M_ oscillations were secondary to changes in neuronal stimulation of the pharynx, responsible for the pumping rate, we have recorded electropharyngeograms (EPGs) in both wild-type N2 and *mcu-1* mutant strains. Figure 2 shows typical EPGs made in both strains, as well as the mean parameters obtained from EPGs performed in 52 wild-type N2 and 77 *mcu-1* mutant worms. Consistent with the changes observed in mitochondrial Ca^2+^ dynamics, we found a significant decrease in the mean frequency in the *mcu-1* mutants, which was due to the increase in the interpumping space (IPI, see panel b). In addition, *mcu-1* mutants showed an increase in the pump duration, which is also consistent with the increase in the width and height of the mitochondrial [Ca^2+^] oscillations. There were no changes in the R/E ratio, which is the ratio of the repolarization and depolarization waves in each pump (R and E waves, see panel a) and is a parameter that reveals possible alterations in the normal muscle contraction/relaxation. Therefore, most of the kinetic changes in the [Ca^2+^]_M_ oscillations could be secondary to changes in the frequency of neuronal stimulation of the pharynx.

### 2.3. Mitochondrial [Ca^2+^] in Control and MCU-Defective Worms under Strong Stimulation

We then reasoned that since MCU opening is triggered by Ca^2+^ with very low affinity, the effect of the absence of MCU would be better revealed after a large stimulation of the pharynx—this was the case. Although the spontaneous Ca^2+^ oscillations showed few changes in the mutants, we could detect an important decrease in the response to carbachol in the *mcu-1* mutants. Figure 3a shows the mean response of pharynx [Ca^2+^]_M_ to carbachol addition in both AQ3055 and *mYCmcu-1* mutants. In control worms, addition of carbachol increased the fluorescence ratio by 25.7 ± 3.0% (*n* = 8), while in the *mYCmcu-1* mutants it only increased by 10.6 ± 1.0% (*n* = 10) (Figure 3b). Therefore, a lack of MCU largely decreased the [Ca^2+^]_M_ peak induced by carbachol, suggesting that under these conditions of intense stimulation, mitochondrial Ca^2+^ uptake takes place mainly via the MCU. In addition, the increase in carbachol-induced [Ca^2+^]_M_ in the *mYCmcu-1* mutants, although smaller, further shows that a different pathway for mitochondrial Ca^2+^ uptake is active in the absence of MCU. The figure also shows that under resting conditions, mitochondrial [Ca^2+^] is smaller in the *mYCmcu-1* mutants than in the controls. However, the difference was not statistically significant (Figure 3c).

## 3. Discussion

In mice, knockout of the MCU protein completely blocked mitochondrial Ca^2+^ uptake triggered by several stimuli that increased cytosolic [Ca^2+^] in different cell types [13]. However, basal aerobic metabolism in muscle was not altered and differences could only be observed under conditions of maximum work, where the MCU knockout was unable to generate maximal power. To explain the very mild phenotype, it has been proposed that additional Ca^2+^ entry pathways could compensate for the lack of MCU in the knockout mouse [19,20,21]. Ca^2+^ could enter mitochondria through reversal of the mitochondrial Na^+^/Ca^2+^ or H^+^/Ca^2+^ exchanger or perhaps through other Ca^2+^ channels which have been localized in mitochondria, such as Ryanodine receptors or TRPC3. However, no significant mitochondrial Ca^2+^ entry was actually measured in the MCU knockouts over a 10–20 min time scale [13] and no changes in heart function were found between the control and knockout mice [14].

In *C. elegans*, the only components of the MCU machinery available are the MCU protein (*mcu-1*) and the regulators MICU1 and EMRE (*micu-1* and *emre-1*). Knockout of the MCU protein in *C. elegans* has been reported to produce no phenotype except for alterations in wound healing [15], in spite of the large reduction in mitochondrial Ca^2+^ uptake observed after wounding. We also found that there is a reduction in mitochondrial Ca^2+^ uptake after strong stimulation with carbachol. However, it was very surprising to find that there were very few changes in the spontaneous Ca^2+^ oscillations of the pharynx. Genotyping of the *mYCmcu-1* mutant worms by PCR and RT-qPCR showed that this strain completely lacks functional MCU expression (Appendix A and RT-qPCR data). Our data therefore indicate that *C. elegans* mitochondria have a pathway different from MCU which is able to carry out fast Ca^2+^ influx to the mitochondria in response to small and more physiological increases in cytosolic [Ca^2+^]. Instead, MCU becomes essential for mitochondrial Ca^2+^ uptake only after major stimulation. Regarding the magnitude of the spontaneous Ca^2+^ oscillations, it is difficult to calculate because we cannot measure the minimum and maximum fluorescence ratio. However, we know that YC3.60 has a Kd of 0.25 µM [22] and addition of carbachol induces a large increase in the ratio over the level of the spontaneous Ca^2+^ oscillations (Figure 1). In wild-type worms, the range of the spontaneous oscillations is only 12 ± 1% (mean ± s.e.m., *n* = 10) of the maximum range obtained after carbachol addition. A resting mitochondrial [Ca^2+^] of around 100 nM suggests that the magnitude of the spontaneous [Ca^2+^]_M_ oscillations is not higher than 200 nM. Our data show that these small changes in mitochondrial [Ca^2+^] occur through a different pathway and do not require the MCU. In fact, this is to be expected given MCU’s low affinity for calcium and taking into account the fact that spontaneous cytosolic [Ca^2+^] oscillations are also of small magnitude [16]. Instead, the large increase in mitochondrial [Ca^2+^] induced by carbachol requires the participation of MCU.

Although the spontaneous Ca^2+^ oscillations remained very similar in the *mcu-1* mutants, they showed an increase in both width and amplitude on the mitochondrial Ca^2+^ peaks, and a reduction in the frequency. The mechanism of these changes is unknown, but EPG data showed also an increase in the pump duration and a decrease in the frequency of pharynx pumping, mainly due to an increase in the interpumping interval. Therefore, the changes in mitochondrial [Ca^2+^] can most probably be attributed to changes in the rate and intensity of neuronal stimulation arriving to the pharynx, perhaps as a consequence of the lack of MCU in the neurons.

In conclusion, our data suggest that mitochondrial Ca^2+^ uptake during low-intensity physiological cell stimulation takes place in the *C. elegans* pharynx through a pathway independent of the MCU. If a similar pathway existed in mammals, it would explain the mild phenotype induced in mice by MCU knockout. In contrast, MCU is essential for transferring Ca^2+^ into the mitochondria during high-intensity stimulation, a mechanism that in some conditions may induce mitochondrial Ca^2+^ overload.

## 4. Materials and Methods

### 4.1. C. elegans Strains and Maintenance

The strains used were as follows: N2 was used as a control; AQ3055, an N2-derived strain expressing mitochondrially targeted yellow cameleon 3.60 (YC3.60) was used as an extrachromosomal array on pharynx, also under the myo-2 promoter (pmyo-2::2mt8::YC3.60) [17]. Its lifespan was not significantly different from that of the N2 strain (data not shown). The CZ19982 strain [15], which corresponds to a mutant *mcu-1* (ju1154), was obtained from the Caenorhabditis Genetics Center. Mutant *mcu-1* strains expressing the mitochondrially targeted YC3.60 (named *mYCmcu-1*) were obtained by crossing AQ3055 worms with CZ19982 ones. Worms were maintained and handled as previously described [23]. Nematode Growth Medium agar plates were seeded with Escherichia coli (OP50). Strains were maintained at 20 °C.

### 4.2. Genotyping and RT-qPCR of the mcu-1 and mYCmcu-1 Mutants

*mcu-1* mutants have no functional mitochondrial Ca^2+^ uniporter, because they contain an 821 base pair (bp) deletion in homozygosis, with breakpoints in exon 2 (CCGCT^CAGTGA) and in intron 5 (TTTTCT^GAAA) [15]. This deletion eliminates the pore region of the MCU proteins, so that it should not be able to transport Ca^2+^ at all. To be sure that full MCU-1 protein was completely absent in the *mcu-1 and mYCmcu-1* mutants, we amplified PCR fragments corresponding to a region of the *mcu-1* gene inside the deleted fragment and also fragments including the deleted region. Primers used for the region inside the deleted region were: forward, CGCCGTGTATGGAACGAGTA; reverse: ATGACTCGATCCGTGTGAGC. They produced a PCR product of 454 bp in wild-type worms and none in the *mcu-1* mutants. Primers used for the region including the deleted region were: forward, CCACAAATGAGGAATGGCCG; reverse: AGCTAACGGGAAGATGCTGG. They produced a PCR product of 1521 bp in the wild-type mutants and only 700 bp in the *mcu-1* mutants. Genomic DNA from 50 adult worms was extracted using the QuickExtract™ DNA Extraction Solution (Epicentre, Lucigen, WI, USA). PCR reactions were performed using MyTaq DNA polymerase (Ecogen, Barcelona, Spain). DNA was initially denatured for 2 min at 95 °C. The PCR cycles were carried out for 30 s at 95 °C, 30 s at 58 °C and 30 s at 72 °C. A total of 35 cycles were carried out. PCR was performed in an Eppendorf Mastercycler personal thermocycler. PCR products were separated by electrophoresis in 1% agarose gels.

Appendix A shows that amplification of a PCR fragment inside the deleted region produced no signal at all in the *mcu-1* mutant, as well as in several *mYCmcu-1* strains. Appendix A shows that amplification of a region including the deleted region produced fragments in agreement with the size of the deletion. Therefore, the *mcu-1* mutant and the *mYCmcu-1* strains completely lack the functional *mcu-1* gene.

RT-qPCR using the primers corresponding to the deleted region of the MCU-1 gene was also carried out to confirm the absence of MCU-1 mRNA. We obtained no amplification at all in the *mYCmcu-1* mutants, while the wild-type worms amplified it with respect to the actin gene (ACT-1) with 2^−ΔCt^ = 0.13 ± 0.07 (mean ± s.e., *n* = 8), with ΔCt = Ct(MCU-1) − Ct(ACT-1). Worms at day 8 of adult life were harvested by centrifugation and washed with water. Total RNA was obtained by freeze/thaw using Trizol (Invitrogen, Waltham, MA, USA) and homogenization using the Minibeadbeater-8, and RNA was then extracted using the RNeasy mini kit (Qiagen, Hilden, Germany), treated with the RNase-Free DNase Set (Qiagen) and then precipitated with ethanol and resuspended in water. RNA concentrations and quality were measured using a NanoDrop spectrophotometer. A reverse transcription reaction was carried out with the iScript™ cDNA Synthesis Kit (BioRad, Richmond, CA, USA) using random primers. RT-qPCR was carried out using the LightCycler 480 PCR system (Roche Applied Science, Penzberg, Germany). Reactions were performed using the SYBR Green Master Mix (Applied Biosystems/Thermo Fisher, Waltham, MA, USA) using the same primers mentioned above for the deleted region. The actin 1 gene (*act-1*) was used as the endogenous control gene. The following program parameters were used for all amplifications: 95 °C for 10 min, followed by 45 cycles at 95 °C for 15 s, 60 °C for 30 s and 72 °C for 30 s, and finally one cycle at 95 °C for 20 min, 65 °C for 1 min and 97 °C for 5 min. Assays were performed using two biological replicates, each consisting of technical cuatriplicates.

### 4.3. Mitochondrial Distribution of the YC3.60 Ca^2+^ Sensor Studied by Confocal Microscopy

To assess the subcellular distribution of the YC3.60 Ca^2+^ sensor in the *mcu-1* mutant crosses, confocal images of the pharynx region were obtained in the presence of the mitochondrial tracker mitotracker deep red, as previously described [17]. Briefly, worms anesthetized with 10 mM tetramisole were imaged on a Leica TCS SP5 confocal microscope. YC3.60 fluorescence was excited at 488 nm, and the fluorescence emitted between 500 and 554 nm was collected. Mitotracker Deep Red FM (Invitrogen, Catalog number M22426) was excited at 644 nm and the fluorescence emitted between 657 and 765 nm was collected. Mitotracker deep red was loaded by incubating the worms for 2 h with 10 µM of the dye in M9 buffer (KH_2_PO_4_ 22 mM; Na_2_HPO_4_ 42 mM, NaCl 86 mM, MgSO_4_ 1 mM). Appendix A shows a clear colocalization of both fluorescences in the region of the anterior bulb, where the mitotracker red is better loaded.

### 4.4. Mitochondrial Calcium Imaging

Pharynx mitochondrial Ca^2+^ measurements were carried out as previously described [17] in worms at day 8 of adult life, which were starved for 4–6 h before the experiments. Then, they were glued (Dermabond Topical Skin Adhesive, Johnson & Johnson, New Brunswick, NJ, USA) on an agar pad (2% agar in M9 buffer) made on a coverslip and 2.3 mM serotonin was added to stimulate pumping. The coverslip containing the glued worm was mounted in a chamber in the stage of a Zeiss Axiovert 200 inverted microscope. Fluorescence was excited at 430 nm using a Cairn monochromator (7-nm bandwidth) and images of the emitted fluorescence obtained with a Zeiss C-apochromat 40 × 1.2 W objective were collected using a 450-nm long pass dichroic mirror and a Cairn Optosplit II emission image splitter to obtain separate images at 480 and 535 nm emission. The splitter contained emission filters DC/ET480/40 m and DC/ET535/30 m, and the dichroic mirror FF509-FDi01-25 × 36 (all from Chroma Technology). Simultaneous 200-ms images at the two emission wavelengths were recorded continuously (2.5-Hz image rate) by a Hamamatsu ORCA-ER camera, in order to obtain 535/480 nm fluorescence ratio images values of a region of interest enclosing the pharynx terminal bulb. Experiments were performed at 20 °C and carried on for 30 min of continuous recording.

Fluorescence was recorded and analyzed using the Metafluor program (Universal Imaging), as previously described [16]. The traces shown are either the 535-nm emission fluorescence, the 480-nm emission fluorescence or the 535 /480-nm fluorescence emission ratio of a region of interest enclosing the pharynx terminal bulb. [Ca^2+^] peaks in the ratio were only considered acceptable when inverted changes at both wavelengths were clearly observed. Fluorescence intensities and ratio changes were then analyzed with a specific algorithm designed to calculate off-line the width at mid-height expressed in seconds, the height obtained as the percent of ratio change and the frequency of all the Ca^2+^ peaks in each experiment. The frequency was measured at each peak as 9 divided by the distance among the peak 4 positions before and the peak 4 positions after. The mean frequency was calculated as the mean of all the individual frequencies higher than 5 peaks/min obtained in worms of a given age and condition.

### 4.5. Electropharyngeogram

To perform electrical recordings of pharynx pumping (electropharyngeogram, EPG), we placed the Nemametrix Screen Chip System on an inverted Zeiss Axiovert 200 microscope equipped with an LD A-Plan 20× objective. In order to minimize interfering electrical noise during the recordings, we used a system grounding shield. Baseline noise was typically between 10 and 40 µV. The experiments were performed at 20 °C.

For each experiment, we picked 100 worms from the culture plate and washed them in 1.5 mL of 0.2 µm filtered M9 buffer +0.1% Tween. Worms were then washed 4× with 0.2 µm filtered M9 buffer, then once in M9 buffer containing 2.3 mM serotonin, and they were finally suspended in 1 mL of M9 buffer containing 2.3 mM serotonin and allowed to settle for 15 min. All the experiments were performed between 15 and 120 min of the initial serotonin exposure. We loaded a NemaMetrix screen chip system (NemaMetrix, Eugene OR; Cat # SK100) with a fresh SC40 screen chip (NemaMetrix, Eugene OR; Cat # SKU: 0002) and added M9 buffer containing 2.3 mM serotonin. After initiating the NemAcquire software and recording the basal power line noise, worms were vacuum-loaded onto the Nemametrix Screen Chip SC40 to start the experiment. The 1-hz high-pass and 50-hz notch filter settings were selected. A 180-s EPG recording was made for each animal, and records from 20–25 animals were obtained for each replicate. Records were analyzed with the NemAnalysis v0.2 software. All the experiments with a frequency of less than 0.1 Hz or pump duration coefficient of variation bigger than 50% were rejected.

### 4.6. Materials

Reagents were obtained from Sigma, Madrid, Spain or Merck, Darmstadt, Germany.

## Figures and Tables

**Figure 1 ijms-21-08622-f001:**
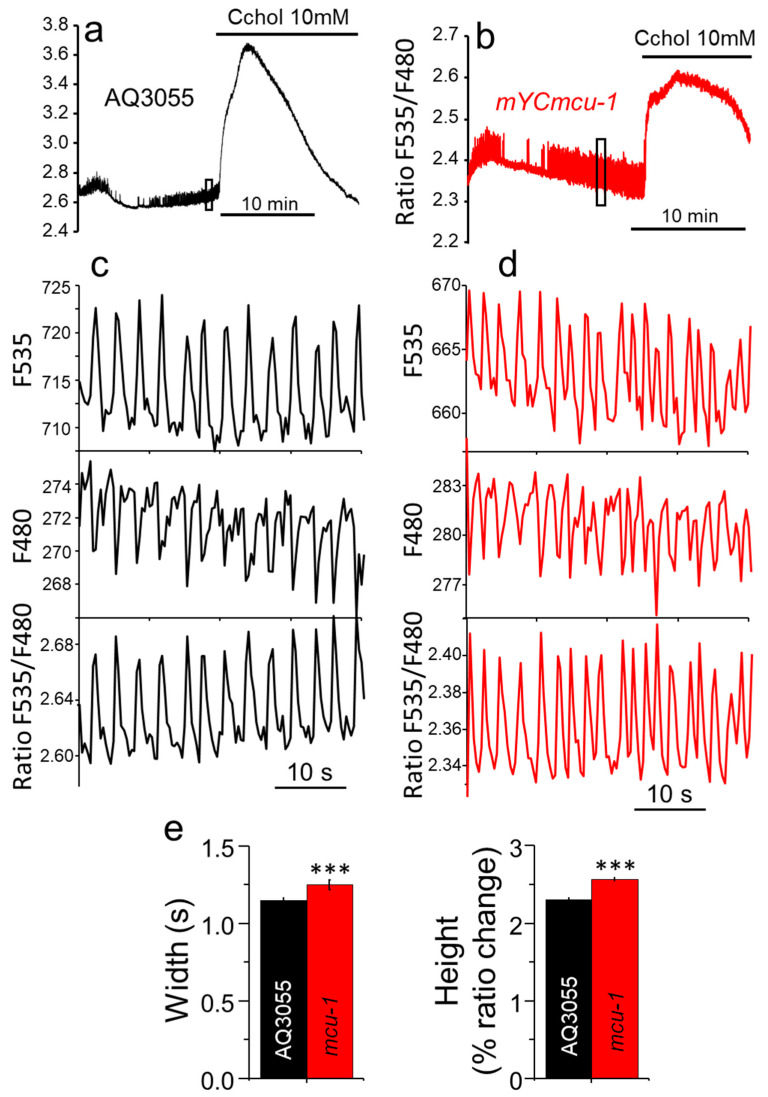
Mitochondrial [Ca^2+^] dynamics in the AQ3055 and *mYCmcu-1* strains. Panels a and b show typical records of fluorescence ratio obtained in the AQ3055 control strain (panel (**a**)) and the *mYCmcu-1* mutant strain, clone #1 (panel (**b**)). When indicated, carbachol (Cchol) was added to the chamber to reach a final concentration of 10 mM. The rectangles in the traces correspond to the region that has been amplified in panels c and d. Panels c and d show the traces corresponding to the two individual F535 and F480 fluorescences (in arbitrary units), and the F535/F480 ratio, both for the AQ3055 strain (panel (**c**)) and for the *mYCmcu-1* strain (panel (**d**)). Panel (**e**) shows the mean peak width and mean peak height obtained in each strain by averaging data obtained from 8 AQ3055 worms (2015 peaks) and 6 mYCmcu-1 worms (1183 peaks). ***, *p* < 0.001, ANOVA test.

**Figure 2 ijms-21-08622-f002:**
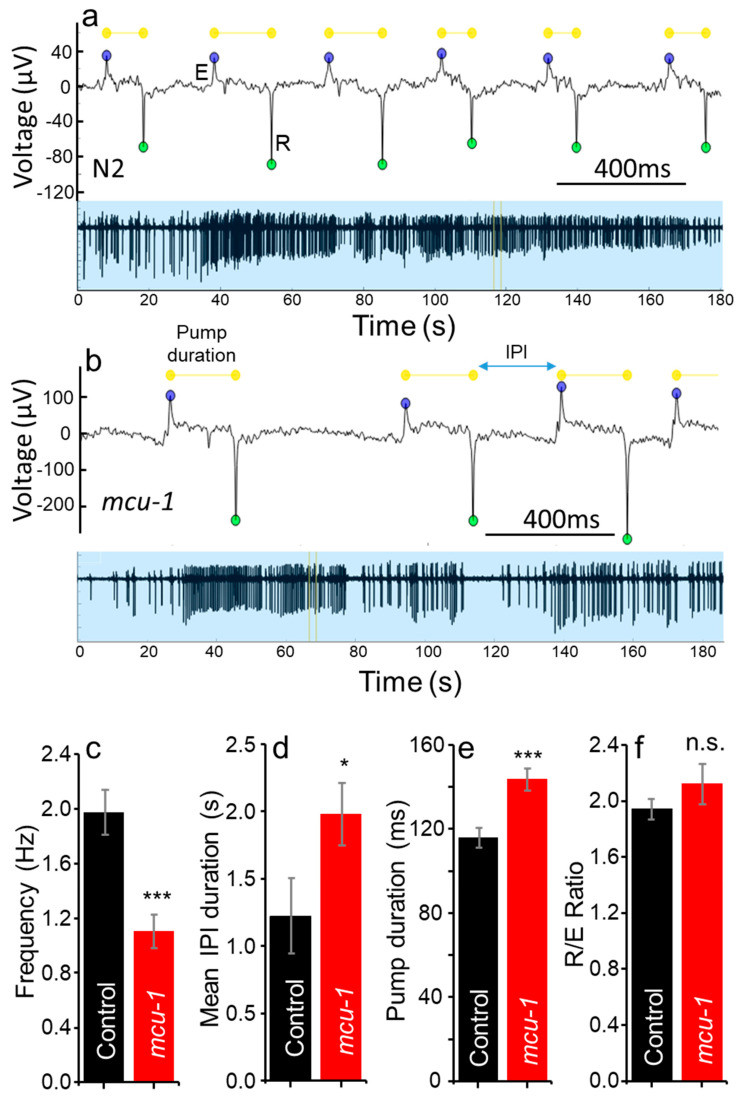
Electropharyngeograms (EPGs) obtained in N2 and *mcu-1* mutants. Panels (**a**,**b**) show typical EPGs obtained either in control N2 worms or in *mcu-1* mutants, respectively. Panel (**c**–**f**) shows the mean values of several parameters obtained from 3-min EPG records of 52 control N2 worms and 77 *mcu-1* mutants. The meaning of the parameters is shown in panels (**a**,**b**) waves E (depolarization) and R (repolarization), pump duration and interpump interval (IPI). ***, *p* < 0.001; *, *p* < 0.05, ANOVA test.

**Figure 3 ijms-21-08622-f003:**
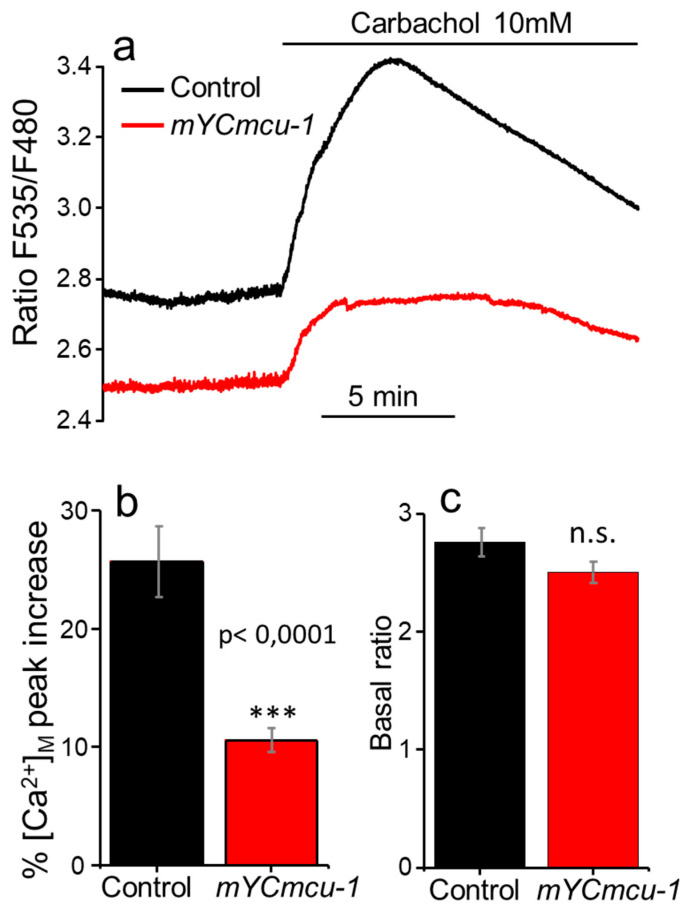
Effect of carbachol on pharynx [Ca^2+^]_M_ in both the control AQ3055 strain and the mutant *mcu-1* strain. Panel (**a**) shows the mean values of fluorescence ratio obtained from 8 experiments performed in control AQ3055 worms and 10 experiments performed in *mcu-1* mutants. Panel (**b**) shows the statistical data on the percentage increase in the ratio induced by carbachol, and panel (**c**) shows the basal ratio in both strains. Comparisons made with ANOVA test. ***, *p* < 0.001.

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
