# Peer review of "Mitochondrial Ca2+ Dynamics in MCU Knockout C. elegans Worms"

_ijms, 2020, doi:10.3390/ijms21228622_

Round 1

Reviewer 1 Report

The work addresses a specific issue, focused on the role of the MCU in mitochondrial Ca2+ homeostasis in baseline conditions and after stimulation. The results obtained in c elegans (wild-type and MCU-knockout) show that this uniporter has a relevant role in Ca2+ entry when cells need a quick response to a stimulus and suggest that other mechanisms are involved under basal conditions, which were however not explored. The work is well designed and structured. The results are clearly described and discussed.  

Author Response

Thank you for the comments.

Reviewer 2 Report

In this paper, authors have examined mitochondrial Ca2+ dynamics in control and MCU-defective C. elegans worms in vivo. This study suggests that MCU is necessary for the fast mitochondrial Ca2+ uptake induced by large cell stimulations but not required at low amplitude mitochondrial Ca2+ oscillations. Major comments
1. Mitochondrial Ca2+ dynamics in MCU knockout C. elegans worms have already been reported with similar findings. mcu-1 null mutants have been reported to display slightly decreased mitochondrial Ca2+ uptake during physiological condition and strongly impaired uptake after wound (Dev. Cell 2014). Although the authors have mentioned this study in the text, the novelty of this paper appears to be limited. In this paper, the authors failed to show experimental shreds of evidence that low amplitude mitochondrial Ca2+ oscillations use a different Ca2+ pathway and are independent of MCU. The authors should also discuss possible independent pathways. 2. Does the basal matrix calcium is different in mcu-1 null mutants? In Fig 1b, baseline calcium recording looks lower in mcu-/- than control.
3. mcu-1 mutants’ characterization is lacking. Although authors have used genotyping, additional MCU knockout expression data should be provided using western or qPCR. Does MCU knockout affect other regulators of mitochondrial calcium signals?
4. Fig 1c and d, Define axis.
5. Does the cytosolic calcium, mitochondrial membrane potential, and metabolism is different in mcu-1 null mutants at low vs high amplitude mitochondrial Ca2+ oscillations?
6. Why Mitochondrial calcium imaging was examined at day 8 of adult worm?
7. Did the authors use a calcium-free buffer for mitochondrial calcium imaging? Please give details in the methods. Minor comments
8. Do not use abbreviations in text for example “& Javier” in the authors’ name 9. The statistic is poorly described. No information on tests and significant value are described. 10. No images are provided for mitochondrial calcium analysis.

Round 2

Reviewer 2 Report

The authors did not address all comments raised by the reviewer, and additional experimental evidence is lacking to improve the paper's quality. The novelty of the data is still limited.